# CAN VLMS REASON THROUGH MULTIPLE VIEWS?

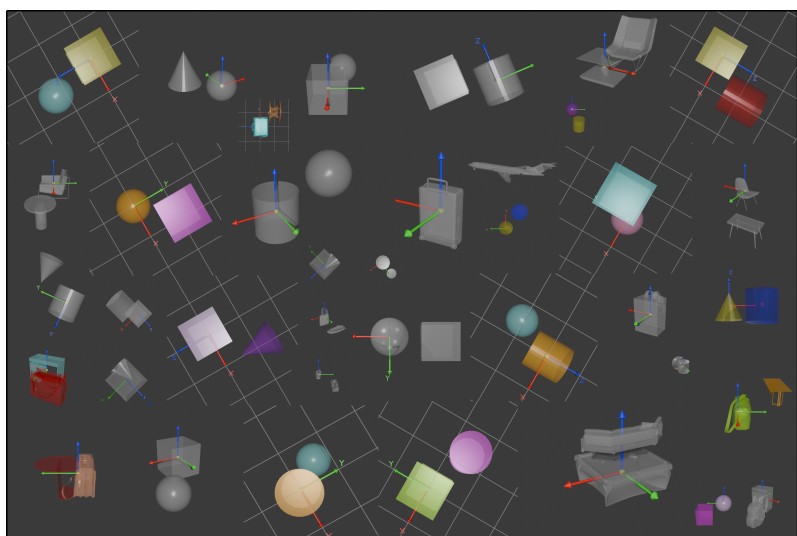

## ABSTRACT

Recent advances in Vision–Language Models (VLMs) have opened new possibilities for complex spatial reasoning. Benchmarks for VLMs largely assess single- or limited-view perception, leaving untested the core ability to *integrate* observations across viewpoints into a coherent 3D understanding. We introduce **MultiView-Bench**, a benchmark expressly designed to evaluate multi-view integration for holistic 3D scene comprehension. MultiView-Bench is paired with a highly *extensible* data-generation pipeline that supports plug-and-play 3D assets (synthetic or real), configurable distractors, and flexible camera positions and orientations, enabling researchers to readily instantiate new datasets by swapping assets or altering viewpoint configurations. Beyond benchmarking, MultiView-Bench serves as a *fundamental diagnostic* that VLMs should pass before being deployed as agents operating 3D software for downstream tasks such as part assembly for mechanical engineering. We evaluate a broad set of frontier VLMs and uncover consistent failure modes: strong performance on 2D planar relations from a single image, but marked difficulty with 3D spatial relations and with aggregating information across views. We further identify biases in VLMs, including handling unconventional axis directions and sensitivity to object colorways and texture variations. Acknowledging these limitations, we propose **ViewNavigator**, a multi-agent framework that actively selects informative viewpoints, perceive, and fuses multi-view evidence through belief-updating. ViewNavigator improves the performances of diverse base models on MultiView-Bench by more than *50%*. MultiView-Bench and its extensible pipeline are designed to equip researchers with a principled testbed for strengthening VLMs' 3D scene understanding, paving the way for more capable VLM-based agents that can support a wide range of downstream 3D tasks. Project webpage: https://hantaozhangrichard.github.io/MultiView-Bench/

# 1 INTRODUCTION

Recent advances in Large Language Models (LLMs) (Brown et al., 2020; OpenAI, 2023; 2024; Touvron et al., 2023) and Vision-Language Models (VLMs) (Radford et al., 2021; Li et al., 2022; Google, 2023; Dai et al., 2023) have demonstrated remarkable progress in complex perceptual and reasoning tasks, including spatial navigation (Mirowski et al., 2018; Du et al., 2023; Yamada et al., 2023) and image understanding (Dosovitskiy et al., 2021). Their strong generalization capabilities, coupled with emergent reasoning skills, make them compelling candidates for *cognitive systems* that integrate perception and strategic planning (Bubeck et al., 2023). When equipped with appropriate tools and scaffolding, such systems have shown promise in robotics control (Brohan et al., 2023; Liu et al., 2023), 3D modeling (Hu et al., 2024; Gu et al., 2025), and image editing (Huang et al., 2024).

However, effectively solving many of these tasks fundamentally depends on the ability to *perceive and reason about scenes from multiple viewpoints* (Edelman, 1998; Bülthoff & Edelman, 1992). Humans naturally perform multi-angle observations to construct coherent mental models of objects, resolving perceptual ambiguities that arise from single viewpoints (Shepard & Metzler, 1971; Palmer, 1999). This ability is crucial when assembling complex objects, where each component must be rotated and inspected from multiple viewpoints to determine how it connects with others. In contrast, a single static image often fails to convey critical structural or relational details necessary for accurate reasoning and manipulation, underscoring the importance of *multi-view perception* in spatial cognition (Marr, 2010; Kosslyn, 1994).

A possible workaround involves geometric representations such as point clouds, meshes, or voxels (Qi et al., 2017a; Wu et al., 2015; Mescheder et al., 2019), which encode precise 3D coordinates and shapes. However, processing such low-level geometric data typically requires specialized encoders (Qi et al., 2017b; Wang et al., 2019) and lacks the broad generalization of LLM/VLM-based approaches. Moreover, these representations diverge from the modality of human visual input, limiting their interpretability for *human-readable reasoning* (Tarr & Bülthoff, 1998).

Current multi-view spatial reasoning benchmarks (Wang et al.; Daxberger et al.; Yin et al., 2025; Zhang et al., 2025) primarily assess egocentric spatial reasoning such as perspective-taking, view-dependent navigation, or dynamic viewpoint transformation. These benchmarks evaluate a model's ability to map pixels from one 2D plane to another or to predict the visual consequences of camera motion without targeting view-invariant (world-centric) spatial understanding and holistic 3D grounding. Such world-centric reasoning ability is a prerequisite for mechanical engineering and operating 3D modeling software (e.g. Blender) where the agent needs to manipulate in global space, independent of where the user is looking.

In this work, we fills this gap by by introducing **MultiView-Bench**, a multi-view spatial reasoning benchmark that explicitly requires the model to integrate multiple viewpoints to reconstruct a static, global mental model of the scene. We ground this by providing a visible, fixed global coordinate system, compelling the model to decouple object positioning from the camera's transient perspective. Our contributions are as follows:

- We introduce **MultiView-Bench**, a diagnostic benchmark for evaluating VLMs' ability to understand the 3D global coordinate system in 3D modeling software and integrate multi-view observations into a coherent world-centric 3D scene understanding. Our benchmark serves as a prerequisite test VLMs should pass before deployment to mechanical engineering and 3D modeling. The main benchmark contains 5 task variants each with 100 tasks where we conduct a systematic evaluation of state-of-the-art VLMs.

- Alongside the benchmark, we provide a flexible **data generation pipeline** that allows researchers to easily extend the dataset with new 3D assets, task variants, and viewpoint configurations. We created 20 more different task variants each with 100 tasks and revealed key failure modes, biases, and limitations in VLMs' multi-view spatial reasoning capabilities through controlled experiments.

- We propose **ViewNavigator**, a multi-agent framework that models perception, planning, and belief-updating. ViewNavigator consistently and significantly enhances the performance of underlying VLMs on MultiView-Bench, demonstrating its potential as a plug-and-play reasoning scaffold for future vision–language systems.

## 2 RELATED WORK

**VLM Benchmarks.** Numerous benchmarks have been developed to evaluate VLM capabilities. Foundational datasets such as ShapeNet (Chang et al., 2015) and ModelNet (Wu et al., 2015) focus on 3D object recognition and reconstruction. CLEVR (Johnson et al., 2017) targets compositional reasoning in synthetic and real-world images. More recent work has extended to spatial reasoning: SpatialRGPT (Cheng et al., 2024) and OmniSpatial (Jia et al., 2025) incorporate perspective-taking and dynamic reasoning, but remain primarily *single-view* in nature.

While multi-view benchmarks exist, they often prioritize different cognitive tasks. InternSpatial (Deng et al., 2025) includes is restricted to rotation estimation. ViewSpatial-Bench (Li et al., 2025), SITE (Wang et al.), SPAR (Zhang et al., 2025) and MindCube (Yin et al., 2025) focus on egocentric–allocentric viewpoints transformations and navigation dynamics that requires sequential feature matching Crucially, these benchmarks do not require the model to integrate multiple concurrent views into a single, coherent global representation. Finally, benchmarks like IR3DBench (Liu et al., 2025) and MM-Spatial (Daxberger et al.) test precise 3D coordinate estimation. However, IR3DBench is restricted to single views which allows multiple plausible configurations to produce the same image and MM–Spatial (Daxberger et al.) relies on explicit depth maps, which hinders scalability compared to pure RGB image reasoning. We summarize the distinctions to those image-only 3D multi-view VLM benchmarks in Table 1.

**3D Spatial Reasoning with LLM/VLM Agents.** Agentic systems leveraging LLMs and VLMs have recently demonstrated impressive performance on 3D tasks, including open-world gaming (Wang et al., 2023; Yao et al., 2023), procedural scene generation (Hu et al., 2024; Huang et al., 2024), and LEGO assembly (Yamada et al., 2024; Pun et al., 2025). In such systems, LLMs often act as *planners*, akin to the prefrontal cortex in the brain (Stokes et al., 2021), while VLMs serve as perceptual modules that transform raw visual inputs into structured descriptions. These pipelines work well when single-image perception suffices (e.g., block-based abstractions in Minecraft), but break down when tasks require precise geometric reasoning over multiple views (Chen et al., 2024; Hong et al., 2023).

While some works attempt to enhance VLM 3D reasoning (Cheng et al., 2024; Chen et al., 2024; Hong et al., 2023), they typically focus on VQA-style setups without extending to real-world applications that require integrated 3D perception and planning. MultiView-Bench is designed precisely to call for the awareness of VLMs' limitations in multi-view integration and to serve as a selection criterion when building VLM-based agents for real-world embodied intelligence as well as for operating 3D software in 3D asset generation and mechanical engineering.

## 3 MOTIVATION

To illustrate the necessity and practical importance of our benchmark, we motivate our study through a real-world furniture part assembly task. In this setting, a collection of labeled components (e.g., legs, table tops, backrests) must be connected and arranged to form a functional piece of furniture. Solving this task naturally demands multi-view perception, 3D spatial reasoning, and common-sense knowledge about how objects are typically used and combined. At a minimum, three core abilities are required:

1. **Part Identification and Semantic Reasoning:** Identifying distinct furniture components and inferring their functionality using combined visual and semantic reasoning.

2. **Assembly Planning:** Formulating a coherent and executable plan by determining the correct assembly sequence and how components interconnect.

3. **Assembly Execution:** Precisely placing each component and executing physical assembly actions.

From experimentation, VLMs have considerable potential to tackle the first two stages, effectively operating as the cognitive system or *brain* of a robotic agent. The subsequent execution can then rely on specialized robotic control modules that function as the robot's *motor system*.

Table 1: Distinctions between existing image-only multi-view VLM benchmarks (Wang et al.; Yin et al., 2025; Zhang et al., 2025) and MultiView-Bench.

| Feature | Existing Multi-View Benchmarks | MultiView-Bench |
|---|---|---|
| **Environment** | **Photorealistic / Real-World Scenes** 
 High visual noise, complex textures, messy lighting. | **Synthetic / Blender-GUI** 
 Clean, schematic geometry with visible global axes/grids. |
| **Reference Frame** | **View-dependent** 
 Relative to the camera (Left, Right, Forward). | **View-invariant** 
 Relative to a fixed global coordinate system. |
| **Reasoning Type** | **Transformation** 
 $View_1 \rightarrow$ Transformation $\rightarrow View_2$ | **Intersection** 
 $View_1 + View_2 + ... \rightarrow$ Global State |
| **Primary Goal** | **End-to-End Performance** 
 Testing general robustness in noisy, real-world environments. | **Diagnostic Prerequisite** 
 "Unit testing" precise spatial logic before deployment. |
| **Domain** | **Navigation & Embodied AI** 
 "Where do I go next?" 
 "What will I see next?" | **Engineering & Manipulation** 
 "Where is this part located exactly?" 
 "Where should I put the next block?" |

While recent LLM-based approaches have shown strong capabilities in environments such as Minecraft or Blender (Yamada et al., 2024; Pun et al., 2025; Wang et al., 2023), where assemblies involve standardized, uniform blocks. Such blocks lend themselves to lossless bounding-box (length, width, height) representation and these LLM-based agents' success largely depends on simple geometric descriptions and mathematical reasoning. However, real-world furniture assembly poses significantly greater complexity due to irregular, non-convex shapes that defy concise, lossless linguistic descriptions. Hence, purely semantic or bounding-box representations are inadequate for precise assembly tasks involving intricate real-world parts.

Figure 1 illustrates typical furniture assembly tasks that explicitly require multi-view perception to comprehend and accurately reason about the spatial configuration of parts. These examples underscore the critical need for robust VLMs capable of integrating information across multiple visual perspectives to build accurate internal 3D understanding.

## 4 MULTIVIEW-BENCH

In this section, we introduce **MultiView-Bench** (Multi-view Benchmark), a foundational evaluation designed to test VLMs' multi-view spatial reasoning capabilities, preparing them for complex real-world tasks like mechanical engineering or 3D scene reconstruction.

### 4.1 SETUP

The core task of MultiView-Bench assesses a VLM's ability to reason about the relative positions of objects within a 3D scene. VLMs must observe scenes from multiple viewpoints to infer spatial relationships accurately (Figure 2). To ensure consistency, we introduce a fixed global coordinate system with clearly marked axes—X (red), Y (green), and Z (blue)—providing a viewpoint-independent frame of reference for spatial descriptions. This coordinate system is created as fixed 3D meshes in Blender so it keeps invariant upon camera change. We added appropriate opacity in objects' materials so that axes would be clearly seen. This is analogous to Blender GUI and this benchmark can be seen as a fundamental test to pass before building any VLM-based agents for 3D model design for manufacturing as it requires VLMs to visually understand 3D coordinate system that is crucial in any 3D modeling software.

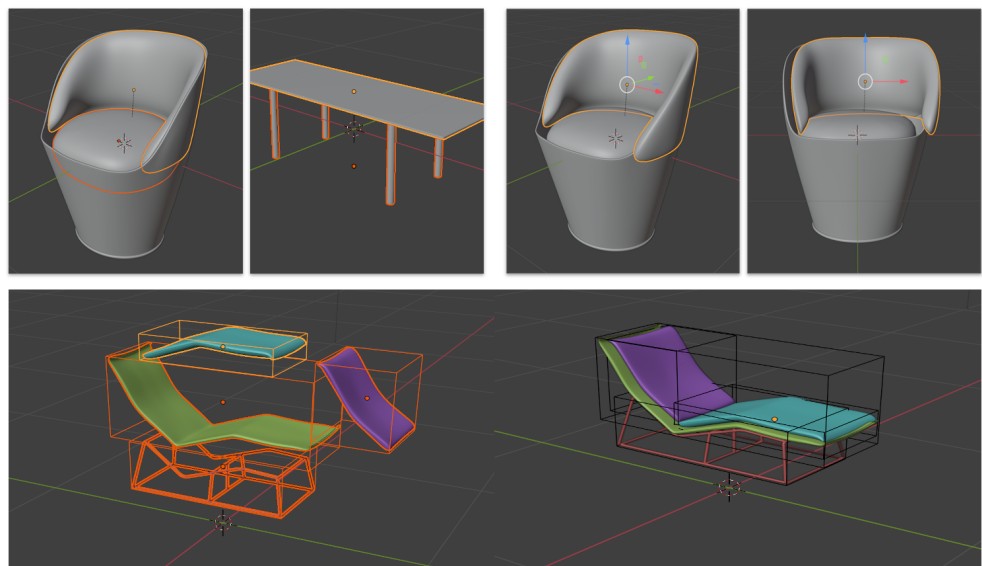

Figure 1: Furniture Assembly Example. Top Left: Many real-world objects do not lend themselves to simple natural language description. The table on the right can be described using fundamental convex shapes and their bounding boxes but the chair on the left has non-convex parts without an analytical expression. Thus it is preferable to include visual information. Top Right: Often times single-view observation leads to visual misconception and does not reveal some alignment issues. The chair looks well-assembled in the view shown on the left but when it turns to the view shown on the right we see the backrest is slightly misplaced in the X-axis. Bottom: Only using the bounding box dimensions, we are unable to assemble furniture that have non-convex parts. In the left instance, the bounding boxes are perfectly aligned but the assembly is problematic. A good assembly example is shown on the right.

After multi-view observations, VLMs must describe the object's relative position to the central object along each axis using the format (±X/0, ±Y/0, ±Z/0), ensuring precise and parsable responses to support large-scale evaluations.

## 4.2 DATASET CREATION

To construct our benchmark dataset, we design a modular pipeline that procedurally generates diverse 3D scenes with controlled variations (Figure 2).

**3D assets.** A wide variety of 3D models are readily available from online repositories. In MultiView-Bench, we used the fundamental geometric object randomly sampled from cube, sphere, cylinder and cone for the synthetic tasks. For real-world objects (3D Real World), we use 3DCoMPaT++ dataset (Slim et al. (2023)) which consists of thousands of real-world objects from different categories like table, chair, airplane and so on. To minimize visual ambiguity in spatial comparisons, all objects are rescaled to share a common bounding box (length, width, height) so that the relative position can be easily inferred from comparison for arbitrary edges or vertices. Beyond the main objects of interest, additional distractor objects can be introduced as confounders to increase scene complexity. The modularity of our pipeline also makes it easy to substitute alternative 3D assets and construct new datasets tailored to specific domains.

**Object placement.** We first fix a central object at the origin and then randomize other objects' positions within the scene while enforcing a minimum and maximum separation distance. This prevents overlaps and ensures objects remain in close proximity. We set a threshold margin for 0 relation by pushing any smaller deviations along an axis to exactly 0. To better analyze model limitations, we also construct controlled task variants where target objects are restricted to simplified spatial layouts. In the DoF=1 variant, objects are placed along a single axis as the central object,

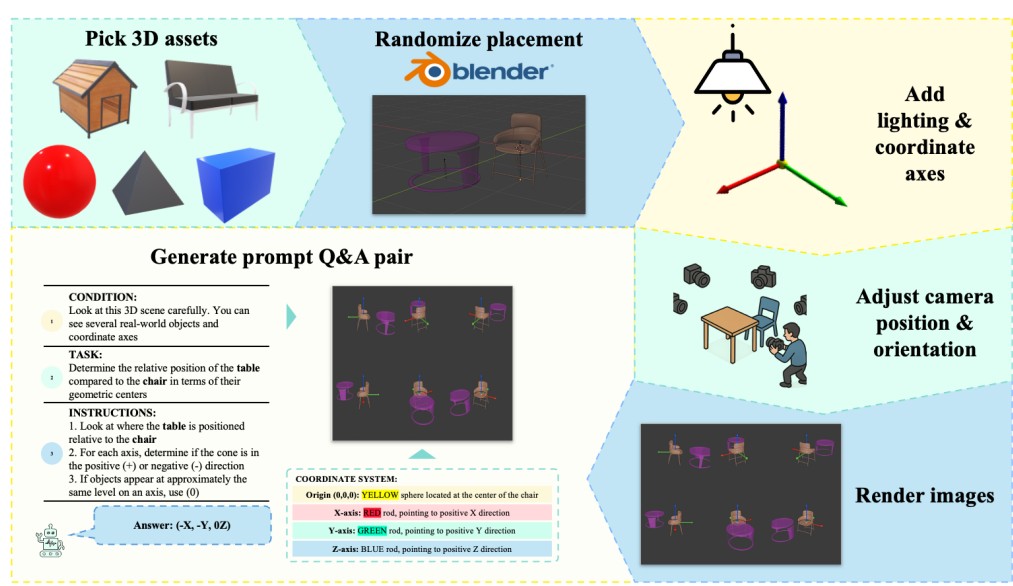

Figure 2: Data generation pipeline of MultiView-Bench: this pipeline allows us to choose 3D assets, manipulate placements, and adjust camera configurations to test different spatial cognitive abilities.

reducing the task to detecting 1D relative relationships. In the DoF=2 variant, objects lie on the same 2D plane as the central object and DoF=3 refers to no constraints of placement in the 3D space.

**Camera viewpoints.** In the main benchmark, we render six viewpoints with uniformly distributed azimuth angles and slight elevations. This configuration guarantees visibility of all three axes. Importantly, the pipeline is not restricted to this setup: arbitrary viewpoint configurations can be specified, enabling analysis of inductive biases in VLMs and the creation of specialized tasks such as egocentric-to-allocentric transformations or spatial navigation through 3D environments. Examples of such extensions to tasks proposed in Jia et al. (2025) and Yin et al. (2025) are provided in the Appendix A.5.

**Rendering.** We render images from each viewpoint. While rendering complex scenes with many textured objects can be computationally expensive, the process is parallelizable across CPU cores, allowing multiple viewpoints to be rendered simultaneously. This design makes large-scale dataset creation both efficient and scalable.

**Q&A generation.** Finally, we generate question–answer pairs automatically. Relative spatial relations are computed directly from Blender's intrinsic coordinate system, ensuring reliable ground-truth supervision.

Overall, this dataset provides a rigorous test of VLMs' ability to integrate multi-view information and reason spatially. Moreover, the extensibility of the pipeline makes it a versatile testbed for probing the limits of 3D reasoning, analyzing inductive biases, and generating tailored training datasets for downstream applications.

## 5 FAILURE AND BIAS ANALYSIS

In this section, we investigate the limitations and biases of off-the-shelf VLMs by conducting detailed case studies on different VLMs.

### 5.1 FAILURE PATTERNS

First we notice that providing only a single isometric view (clearly displaying all three axes) significantly reduced accuracy for both models (Figure 3, 3D Single View), underscoring the essential

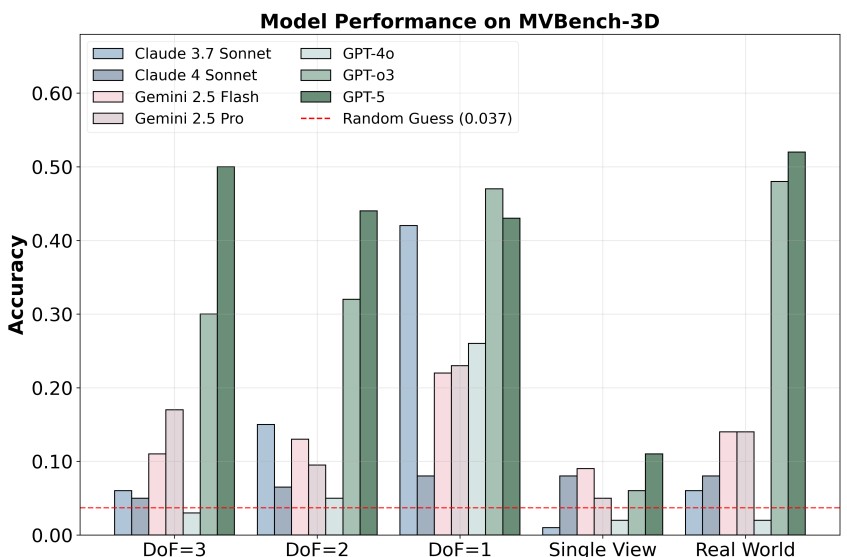

Figure 3: Model Performance on MultiView-Bench: we systematically evaluated the performance of 7 leading VLMs. We found that GPT-5 has the best overall performance while models like Claude series and GPT-4o can hardly beat random chance (red dashed line) in harder tasks 3D DoF=3 and 3D Real World. As we decrease the DoF, models tend to have higher accuracy with a great leap as we decrease the DoF to 1. Surprisingly, Claude 3.7 Sonnet beat its successor significantly in low DoF synthetic tasks and reached near GPT-5 level when DoF=1. We also show that single isometric view is not enough for solving this task as all models' performances are around random chance. Full evaluation results are presented in Appendix A.1. The random chance is computed by $\frac{1}{3^3}$ to account for the 3 choices for each of the 3 axes.

role of multi-view observations. Single-view conditions resulted in substantial information loss, demonstrating that accurate spatial reasoning fundamentally requires multi-angle perception.

To better understand the sources of these failures, we decomposed the spatial reasoning task into intermediate sub-steps analogous to human reasoning processes:

- **Step 1: Object Identification**: Recognizing and distinguishing individual objects.

- **Step 2: 2D Spatial Relation**: Articulating the relative object positions within a given 2D viewpoint.

- **Step 3: Axis Direction Identification**: Interpreting and clearly describing the directionality of each axis within a 3D scene.

- **Step 4: 3D Position Translation**: Translating observed 2D spatial relationships into explicit 3D positional descriptions.

We performed some web trials and revealed that VLMs primarily struggle with accurately identifying and articulating the directionality of axes within the 3D context (Step 3), as illustrated in Figure 4. Models consistently exhibited difficulty in expressing axis directions using clear and unambiguous 3D spatial language.

Motivated by this observation, we further explored whether VLM performance could be enhanced by decomposing the task into simpler 2D views. We designed a variant of the MultiView-Bench task (2D) utilizing three canonical (front, side, top) views, as shown in Figure 8, each emphasizing one 2D plane (XZ, YZ, XY) and clearly displaying only two axes per view. Models were tested under two configurations: first, providing all three canonical views simultaneously to produce a single integrated 3D answer (Single-agent), and second, utilizing a multi-agent approach wherein each agent independently assessed one canonical view, with the final 3D answer obtained by straightforward integration of individual responses (resolving inconsistencies by random selection).

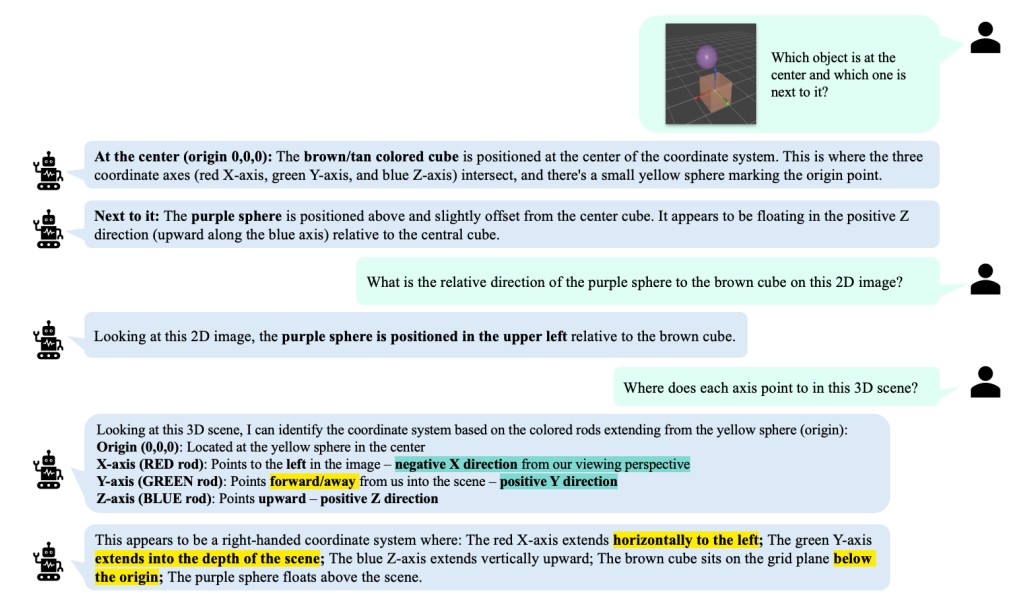

Figure 4: Example conversation with VLM: VLM did well in 2D reasoning in the first 2 steps but failed to reason about the 3D scene in Step 3. Wrong reasoning traces are highlighted in yellow. Signs of inductive bias on the 3D coordinate system are highlighted in cyan.

Results from these experiments, depicted in Figure 5, demonstrate notable improvements by simplifying 3D reasoning tasks into 2D multi-view reasoning tasks and even further improvements by decomposing multi-view 2D tasks into single-view sub-tasks. These results validate that VLMs' struggle both at 3D perception and multi-view integration.

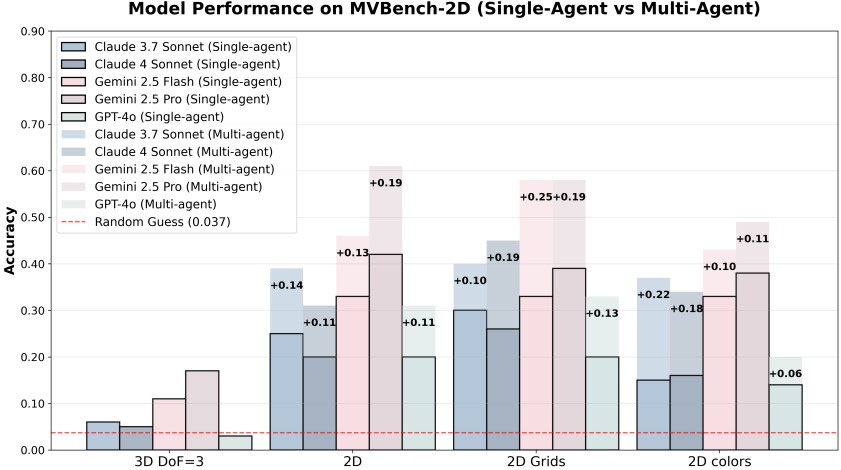

Figure 5: By simplifying the task into 2D multi-view task, VLMs demonstrate significantly higher performance. Decomposing 2D multi-view task into a multi-agent 2D single-view task can further improve VLMs' performances by a great margin. Adding visual aids like grids and distinct colors enhances the performance of Claude series but shows limited help to Gemini series and GPT-4o.

## 5.2 BIAS DISCOVERY

**Visual Enhancements.** We further explore how visual enhancements, such as distinct color schemes and grids, influence VLMs' 3D perception and spatial reasoning performance. Specifically, we evaluated VLMs under conditions of randomized color assignment, fixed color combinations, 3D

scenes with integrated 2D grid planes, and purely 2D views with grid overlays. Previous research indicates that structured visual aids can improve VLM performance on visual reasoning tasks like counting and scene comprehension (Izadi et al., 2025). Our experiments confirm that visual structures indeed boost performance for models like Claude series; however, surprisingly, Gemini 2.5 Pro's performance declines under these conditions. Additionally, we observed distinct color biases among different models, with each showing preferential responses to particular color combinations, underscoring inherent perceptual biases (Figure 6).

**Coordinate Rotation.** Upon closer examination of model responses, we identified a notable pattern: models frequently disregarded explicitly depicted coordinate directions, defaulting instead to reasoning based on the conventional right-handed coordinate system (Figure 4). We hypothesize that this behavior reflects a strong inductive bias derived from extensive exposure to standard coordinate conventions during pretraining. To rigorously investigate this bias, we conducted rotation experiments using the canonical 2D view task. In these experiments, axes directions were deliberately rotated away from conventional orthogonal orientations (such as 90°, 180°, etc.) to non-standard angles. Results shown in Figure 6 clearly showed significant performance degradation under these unconventional orientations, confirming that VLMs heavily rely on learned coordinate-system priors. These findings highlight the critical need to address such inherent biases to enhance the robustness and generalization capabilities of VLMs.

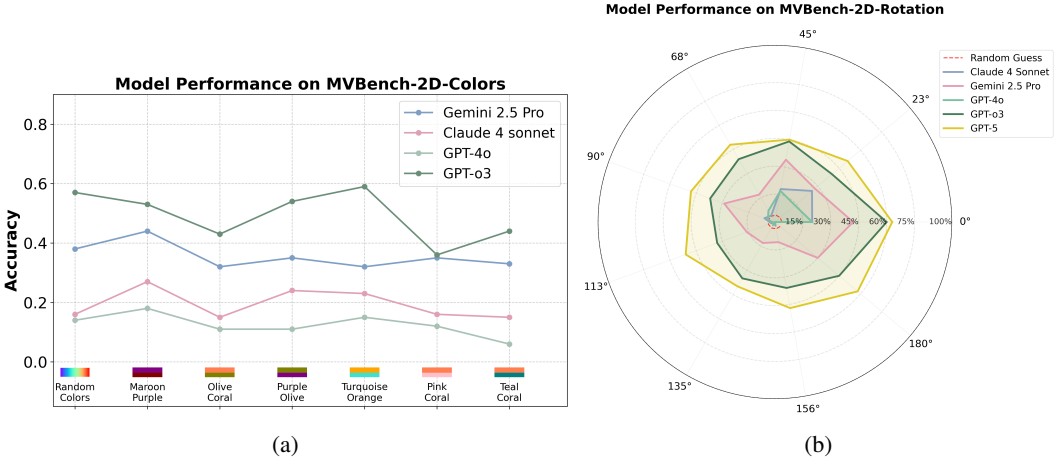

(a)                                                                 (b)

Figure 6: (a) VLMs exhibit fluctuating performance across different color combinations. Notably, these fluctuations diverge from human perception: the models perform better on the less distinguishable Maroon/Purple pair, yet worse on the more distinct Olive/Coral pair. (b) VLMs' performances degrade under unconventional axis orientations.

## 6 VIEWNAVIGATOR

In this section, we introduce ViewNavigator, a multi-agent system designed to actively reason about spatial relationships between objects within a 3D environment.
Our agent architecture integrates a VLM and a LLM in a closed-loop manner without requiring post-training or external geometry-based image analysis. The LLM strategically plans the next move, deciding the viewpoint to look at. The VLM processes visual inputs from one viewpoint and its jittered viewpoints each time. The probablistic belief module (details in Appendix A.3) integrates feedback from VLM to maintain a memory of the trajectory and belief state, which the LLM retrieves to guide future actions. The LLM emits the final answer if it is confident enough.

### 6.1 BELIEF MODULE

The agent maintains a probabilistic belief over spatial directions for each axis using a Dirichlet distribution. After sampling multiple jittered views, the VLM's categorical votes are aggregated into smoothed proportions. To guard against unstable or noisy predictions, we discount the evidence size

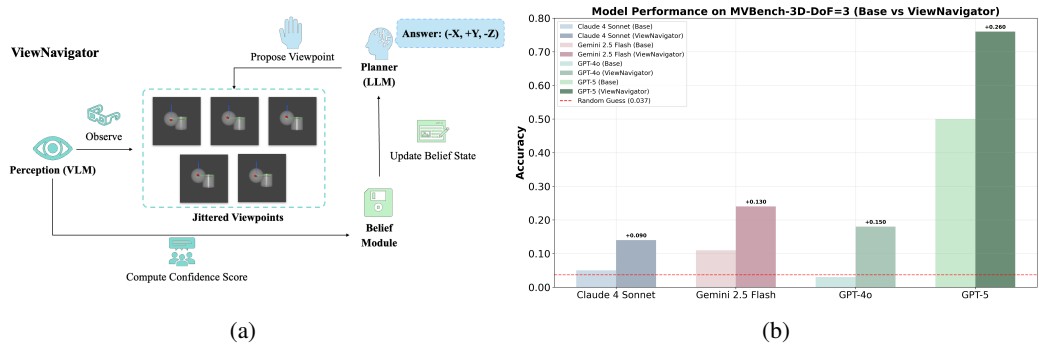

(a)                                            (b)

Figure 7: (a) ViewNavigator workflow. (b) ViewNavigator framework significantly enhances various base models' performances on the 3D DoF=3 tasks.

with a confidence score, computed either from a Wilson lower bound on the majority vote or from the entropy of the vote distribution. The resulting effective evidence determines how strongly the current observations update the belief state. This iterative procedure ensures that stable, consistent clusters strengthen the belief, while ambiguous clusters contribute more cautiously. Full details can be referred to A.3.

## 6.2 ACTIVE VIEW SELECTION AND AGGREGATION

The LLM planner actively proposes the next best camera viewpoint based on the current belief state and previous view history, aiming to maximize information gain and reduce uncertainty. After each belief update, the agent checks if the posterior probability of the dominant class on each axis exceeds a confidence threshold $\tau$ (e.g., $\tau = 0.9$) and if sufficient evidence concentration is reached (e.g., total evidence $\sum_s \alpha_{A,s} \geq \kappa_{\min}$). If these criteria are met for all axes, the agent terminates the viewpoint exploration and outputs the final prediction:

$$\text{prediction} = \arg \max_{s \in \{+, 0, -\}} \frac{\alpha_{A,s}}{\sum_t \alpha_{A,t}}, \quad \forall A \in \{X, Y, Z\}. \tag{1}$$

ViewNavigator significantly enhances the performances across diverse base models by a large margin (Figure 7). Detailed configurations and prompts for ViewNavigator is presented in Appendix A.4

## 7 CONCLUSION

In this work, we presented **MultiView-Bench**, a diagnostic benchmark specifically designed to test the ability of VLMs to understand 3D global coordinate system and integrate multi-view observations into a coherent 3D scene understanding. Alongside the benchmark, we introduced an extensible **data-generation pipeline** that allows researchers to readily construct new datasets and a multi-agent framework **ViewNavigator** that significantly improves the performance of diverse base models on MultiView-Bench. Our systematic evaluation of leading VLMs on MultiView-Bench revealed fundamental limitations: while these models excel at recognizing 2D planar relations from single images, they struggle with integrating information across multiple views, interpreting 3D spatial relations, and generalizing under unconventional axes or texture variations. Taken together, MultiView-Bench, its extensible pipeline, and ViewNavigator form both a diagnostic tool and a stepping stone toward more powerful VLM-based agents. This benchmark is designed to raise awareness of the limitations of current VLMs in multi-view integration and horizontally benchmark and track improvements of VLMs. Our benchmark also serves as a selection standard for base model when building VLM-based 3D-reasoning agents for mechanical engineering or operating 3D modeling software GUI. We hope that future research builds on this foundation to equip VLMs with the spatial understanding necessary for diverse downstream 3D tasks such as part assembly, scene editing, and 3D assets creation.

ACKNOWLEDGMENTS

I would like to express my sincere gratitude to Yujing Zhang for her invaluable contribution in stylizing the graphics used in this paper.

REPRODUCIBILITY STATEMENT

We ensure reproducibility of our experiments by reporting the exact hyperparameters, experimental setup, and prompts. For all VLM and LLM API calls (except GPT-o3 and GPT-5 where the temperature is fixed), we use `temperature = 1.0` and `max_tokens = 4096`. For model evaluations, we used 100 tasks per task variant and repeated each experiment three times, reporting the averaged results. For ViewNavigator, due to the cost and time, we use 50 tasks per task variant and repeated three times as well. Full details of the experiments setup and the exact prompts for each task are provided in Appendix A.4.

LLM USAGE STATEMENT

LLM is used to polish a few sentences in the paper and format some mathematical expressions in LaTeX.

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

# A  APPENDIX

## A.1  MODEL EVALUATIONS IN ALL MULTIVIEW-BENCH TASKS

Table 2: Model Performance on MultiView-Bench

| Tasks/Models | Claude 3.7 Sonnet | Claude 4 Sonnet | Gemini 2.5 Flash | Gemini 2.5 Pro | GPT-4o | GPT-5 | GPT-o3 |
|---|---|---|---|---|---|---|---|
| 3D DoF=3 | 0.060 | 0.050 | 0.110 | 0.170 | 0.030 | **0.500** | 0.300 |
| 3D DoF=2 | 0.150 | 0.065 | 0.130 | 0.095 | 0.050 | **0.440** | 0.320 |
| 3D DoF=1 | 0.420 | 0.080 | 0.220 | 0.230 | 0.260 | 0.430 | **0.470** |
| 3D Single View | 0.010 | 0.080 | 0.090 | 0.050 | 0.020 | **0.110** | 0.060 |
| 3D Real World | 0.060 | 0.080 | 0.140 | 0.140 | 0.020 | **0.520** | 0.480 |
| 2D Three Views (Rotation 0) | 0.250 | 0.200 | 0.330 | 0.420 | 0.200 | **0.630** | 0.550 |
| 2D Three Views multiagent (Rotation 0) | 0.390 | 0.310 | 0.460 | 0.610 | 0.310 | **0.660** | 0.560 |
| 2D Three Views Grids | 0.300 | 0.260 | 0.330 | 0.390 | 0.200 | **0.690** | 0.540 |
| 2D Three Views Grids multiagent | 0.400 | 0.450 | **0.580** | **0.580** | 0.330 | **0.580** | 0.570 |
| 2D Three Views Colors (Random) | 0.150 | 0.160 | 0.330 | 0.380 | 0.140 | **0.640** | 0.560 |
| 2D Three Views Colors multiagent (Random) | 0.370 | 0.340 | 0.430 | 0.490 | 0.200 | **0.700** | 0.420 |
| 2D Three Views Colors (Maroon+Purple) | 0.190 | 0.270 | 0.330 | 0.440 | 0.180 | **0.680** | 0.590 |
| 2D Three Views Colors (Turquoise+Orange) | 0.210 | 0.230 | 0.300 | 0.320 | 0.150 | **0.710** | 0.530 |
| 2D Three Views Colors (Purple+Olive) | 0.230 | 0.240 | 0.320 | 0.350 | 0.110 | **0.660** | 0.540 |
| 2D Three Views Colors (Teal+Coral) | 0.150 | 0.150 | 0.200 | 0.330 | 0.060 | **0.530** | 0.430 |
| 2D Three Views Colors (Olive+Coral) | 0.200 | 0.150 | 0.200 | 0.320 | 0.110 | **0.600** | 0.440 |
| 2D Three Views Colors (Pink+Coral) | 0.180 | 0.160 | 0.130 | 0.350 | 0.120 | **0.580** | 0.360 |
| 2D Three Views Rotation (23°) | 0.210 | 0.260 | 0.310 | 0.280 | 0.140 | **0.510** | 0.400 |
| 2D Three Views Rotation (45°) | 0.240 | 0.180 | 0.310 | 0.340 | 0.170 | **0.450** | 0.440 |
| 2D Three Views Rotation (68°) | 0.090 | 0.040 | 0.290 | 0.170 | 0.070 | **0.480** | 0.390 |
| 2D Three Views Rotation (90°) | 0.060 | 0.060 | 0.130 | 0.290 | 0.040 | **0.480** | 0.370 |
| 2D Three Views Rotation (113°) | 0.030 | 0.020 | 0.200 | 0.160 | 0.010 | **0.510** | 0.330 |
| 2D Three Views Rotation (135°) | 0.070 | 0.000 | 0.160 | 0.130 | 0.020 | **0.400** | 0.350 |
| 2D Three Views Rotation (156°) | 0.030 | 0.020 | 0.170 | 0.110 | 0.010 | **0.470** | 0.360 |
| 2D Three Views Rotation (180°) | 0.000 | 0.000 | 0.120 | 0.300 | 0.000 | **0.580** | 0.450 |

## A.2  MORE EXAMPLE TASKS

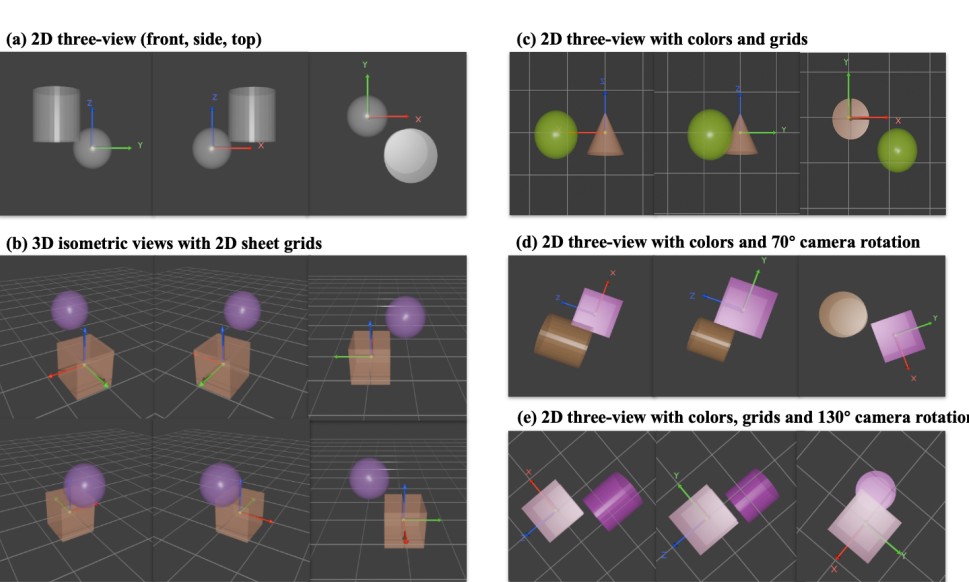

**(a) 2D three-view (front, side, top)**

**(c) 2D three-view with colors and grids**

**(b) 3D isometric views with 2D sheet grids**

**(d) 2D three-view with colors and 70° camera rotation**

**(e) 2D three-view with colors, grids and 130° camera rotation**

Figure 8: More Example Tasks

## A.3 BELIEF STATE AND UPDATE

The agent maintains a probabilistic belief state over spatial directions (+, 0, -) for each axis independently. Specifically, we model this belief as a Dirichlet distribution parameterized by vector $\boldsymbol{\alpha}_A = [\alpha_{A,+}, \alpha_{A,0}, \alpha_{A,-}]$ for each axis $A \in \{X, Y, Z\}$. Initially, each axis is given an uniform prior: $\boldsymbol{\alpha}_A = [1, 1, 1]$.

Upon selecting a camera viewpoint, the agent captures multiple images using micro-jitters (small perturbations around a base viewpoint) to assess stability in the VLM's answers, which is used as a confidence score. For each jittered viewpoint, the VLM returns a categorical judgment $(+, 0, -)$ independently for each axis, resulting in vote counts $k_{A,+}, k_{A,0}, k_{A,-}$ from a set of $n$ images.

To update the belief, we first compute smoothed proportions:

$$\hat{p}_{A,s} = \frac{k_{A,s} + \lambda}{n + 3\lambda}, \quad s \in \{+, 0, -\} \tag{2}$$

where $\lambda$ is a smoothing constant (default $\lambda = 1$) to avoid over-confidence for small n.

These proportions represent the directional preference of the cluster, while the confidence score discounts clusters that show large variability under micro-jitters.

We propose two methods to compute the confidence score: **Wilson Lower Bound Score** and **Relative Entropy Score**, both of which achieved comparable performance.

**Wilson Lower Bound Score.** Given that the majority label among the $n$ answers occurs $k_{\max}$ times, the empirical majority proportion is $\hat{p} = k_{\max}/n$. The Wilson score interval offers a conservative estimate of the true binomial proportion, particularly robust for small $n$ or when proportions are near 0 or 1. The 95% Wilson lower bound is computed as:

$$\text{LB} = \frac{\hat{p} + z^2/(2n) - z\sqrt{\frac{\hat{p}(1-\hat{p})}{n} + \frac{z^2}{4n^2}}}{1 + z^2/n}, \quad z = 1.96. \tag{3}$$

We map this to a conservative confidence score relative to a random baseline (uniform guess = $1/3$):

$$\omega_A = \left( \frac{\max(\text{LB}, 1/3) - 1/3}{2/3} \right)^\gamma, \tag{4}$$

where $\gamma \in [1, 2]$ controls sensitivity.

**Relative Entropy Score.** Let $H(\hat{\mathbf{p}}_A) = -\sum_s \hat{p}_{A,s} \log \hat{p}_{A,s}$ be the entropy of the smoothed vote distribution and $H_{\max} = \log 3$ its maximum for three equally likely outcomes. The normalized entropy gap from uniform is:

$$\omega_A = \left( 1 - \frac{H(\hat{\mathbf{p}}_A)}{H_{\max}} \right)^\gamma, \tag{5}$$

with $\gamma$ again controlling sensitivity.

These two methods prevent overconfidence when a cluster's votes are unstable, rewarding highly peaked vote distributions and penalizing near-uniform ones.

The effective evidence size is then:

$$n_{\text{eff},A} = n \cdot \omega_A. \tag{6}$$

**Belief Update.** The smoothed proportions $\hat{p}_{A,s}$ are scaled by $n_{\text{eff},A}$ to yield soft counts:

$$\Delta\alpha_{A,s} = n_{\text{eff},A} \cdot \hat{p}_{A,s}, \quad \forall s \in \{+, 0, -\} \tag{7}$$

These are added to the Dirichlet parameters to yield the new belief:

$$\alpha_{A,s} \leftarrow \alpha_{A,s} + \Delta\alpha_{A,s} \tag{8}$$

This belief is updated iteratively over successive jittered view clusters.

**Active View Selection and Aggregation**  The LLM planner actively proposes the next best camera viewpoint based on the current belief state and previous view history, aiming to maximize information gain and reduce uncertainty. After each belief update, the agent checks if the posterior probability of the dominant class on each axis exceeds a confidence threshold $\tau$ and if sufficient evidence concentration is reached (e.g., total evidence $\sum_s \alpha_{A,s} \geq \kappa_{\min}$). If these criteria are met for all axes, the agent terminates the viewpoint exploration and outputs the final prediction:

$$\text{prediction} = \arg \max_{s \in \{+,0,-\}} \frac{\alpha_{A,s}}{\sum_t \alpha_{A,t}}, \quad \forall A \in \{X, Y, Z\}. \tag{9}$$

## A.4 Experiments Setup

All experiments are conducted using our proposed **MultiView-Bench** benchmark. The 3D scenes and corresponding multi-view images are procedurally generated using Blender. Each scene is constructed with a fixed global coordinate system, represented by colored axes (X: red, Y: green, Z: blue), to provide a consistent frame of reference across all viewpoints. The generation pipeline allows for the use of various 3D assets, randomized object placements, and configurable camera positions, as detailed in Section 4 of the main paper.

### A.4.1 Hyperparameters

**VLM API Calls (Single-Agent & Multi-Agent)**  For models other than GPT-o3 and GPT-5:

- `temperature = 1.0`
- `max_tokens = 4096`

**ViewNavigator Framework**  The ViewNavigator agentic framework was configured as follows:

- `max_steps = 10` (Maximum number of viewpoints the agent can select)
- `r_az = 5` (Radius in degrees for azimuthal jitter)
- `r_el = 5` (Radius in degrees for elevation jitter)
- `tau = 0.6` (Confidence threshold $\tau$ for the belief state using the Wilson Lower Bound Score)
- `jitter_size = 5` (Number of jittered images per viewpoint)

### A.4.2 PROMPTS

The exact prompts used in our experiments are provided below. Placeholders such as {central_obj_type} are filled dynamically during data generation. Labels such as % VLM SYSTEM PROMPT are included only for readability in the paper.

**Prompt for MultiView-Bench-3D and MultiView-Bench-2D Tasks (Single-agent):**

```
Look at this 3D scene carefully from different viewpoints. You can
see several
geometric objects and coordinate axes.

COORDINATE SYSTEM:
- X-axis: RED rod, pointing to positive X direction
- Y-axis: GREEN rod, pointing to positive Y direction
- Z-axis: BLUE rod, pointing to positive Z direction
- Origin (0,0,0): YELLOW sphere, located at the center of the
{central_obj_type}

TASK:
Determine the relative position of the {sampled_obj_type} compared
to the
{central_obj_type} in terms of their geometric centers.

INSTRUCTIONS:
1. Look at where the {sampled_obj_type} is positioned relative to
the {central_obj_type}
2. For each axis, determine if the {sampled_obj_type} is in the
positive (+) or
   negative (-) direction using the coordinate system shown in the
   images.
3. If objects appear at approximately the same level on an axis,
use (0)

ANSWER FORMAT:
Respond with exactly this format: <answer>(±X, ±Y, ±Z)</answer>
Examples: <answer>(+X, -Y, +Z)</answer> or <answer>(-X, 0Y,
-Z)</answer>
or <answer>(0X, +Y, 0Z)</answer>

What is the relative position of the {sampled_obj_type} to the
{central_obj_type}?
```

**Prompt for MultiView-Bench-2D Tasks (Multi-agent):**

```
% Prompt for Front View (XZ plane)
Look at this Front View (XZ plane) carefully. You can see several
geometric objects and coordinate axes.

VIEW DESCRIPTION:
This is the Front View (XZ plane), looking along the Y-axis.

COORDINATE SYSTEM:
- X-axis: RED rod, pointing to positive X direction
- Z-axis: BLUE rod, pointing to positive Z direction
- Origin (0,0,0): YELLOW sphere, located at the center of the
{central_obj_type}

TASK:
Determine the relative position of the {sampled_obj_type} compared
to the {central_obj_type} in terms of their geometric centers,
focusing only on the X and Z axes visible in this view.

INSTRUCTIONS:
1. Look at where the {sampled_obj_type} is positioned relative to
the {central_obj_type}
2. For each visible axis (X, Z), determine if the
{sampled_obj_type} is in the positive (+) or negative (-) direction
using the coordinate system shown in the image.
3. If objects appear at approximately the same level on an axis,
use (0)

ANSWER FORMAT:
Respond with exactly this format for the X and Z axes: <answer>(±X,
±Z)</answer>
Examples: <answer>(+X, -Z)</answer> or <answer>(0X, +Z)</answer>

What is the relative position of the {sampled_obj_type} to the
{central_obj_type} in the X and Z axes?
```

```
% Prompt for Side View (YZ plane)
Look at this Side View (YZ plane) carefully. You can see several
geometric objects and coordinate axes.

VIEW DESCRIPTION:
This is the Side View (YZ plane), looking along the X-axis.

COORDINATE SYSTEM:
- Y-axis: GREEN rod, pointing to positive Y direction
- Z-axis: BLUE rod, pointing to positive Z direction
- Origin (0,0,0): YELLOW sphere, located at the center of the
{central_obj_type}

TASK:
Determine the relative position of the {sampled_obj_type} compared
to the {central_obj_type} in terms of their geometric centers,
focusing only on the Y and Z axes visible in this view.

INSTRUCTIONS:
1. Look at where the {sampled_obj_type} is positioned relative to
the {central_obj_type}
2. For each visible axis (Y, Z), determine if the
{sampled_obj_type} is in the positive (+) or negative (-) direction
using the coordinate system shown in the image.
3. If objects appear at approximately the same level on an axis,
use (0)

ANSWER FORMAT:
Respond with exactly this format for the Y and Z axes: <answer>(±Y,
±Z)</answer>
Examples: <answer>(+Y, -Z)</answer> or <answer>(0Y, +Z)</answer>

What is the relative position of the {sampled_obj_type} to the
{central_obj_type} in the Y and Z axes?"
```

```
% Prompt for Top View (XY plane)
Look at this Top View (XY plane) carefully. You can see several
geometric objects and coordinate axes.

VIEW DESCRIPTION:
This is the Top View (XY plane), looking along the Z-axis from
above.

COORDINATE SYSTEM:
- X-axis: RED rod, pointing to positive X direction
- Y-axis: GREEN rod, pointing to positive Y direction
- Origin (0,0,0): YELLOW sphere, located at the center of the
{central_obj_type}

TASK:
Determine the relative position of the {sampled_obj_type} compared
to the {central_obj_type} in terms of their geometric centers,
focusing only on the X and Y axes visible in this view.

INSTRUCTIONS:
1. Look at where the {sampled_obj_type} is positioned relative to
the {central_obj_type}
2. For each visible axis (X, Y), determine if the
{sampled_obj_type} is in the positive (+) or negative (-) direction
using the coordinate system shown in the image.
3. If objects appear at approximately the same level on an axis,
use (0)

ANSWER FORMAT:
Respond with exactly this format for the X and Y axes: <answer>(±X,
±Y)</answer>
Examples: <answer>(+X, -Y)</answer> or <answer>(0X, +Y)</answer>

What is the relative position of the {sampled_obj_type} to the
{central_obj_type} in the X and Y axes?
```

**Prompts for ViewNavigator:**

VLM PERCEPTION MODULE PROMPTS

```
% VLM SYSTEM PROMPT
You are a precise vision judge. The image shows colored world axes:

COORDINATE SYSTEM:
- X-axis: RED rod, pointing to positive X direction
- Y-axis: GREEN rod, pointing to positive Y direction
- Z-axis: BLUE rod, pointing to position Z direction
- Origin (0,0,0): YELLOW sphere, located at the center of the
{central_object}
- Both CENTRAL and TARGET objects have the same scale in X, Y, Z
dimensions

TASK:
Determine the relative position of the {target_object} compared to
the
{central_object} in terms of their geometric centers.

INSTRUCTIONS:
1. Only focus on axis {axis} for this view and only give answer for
these axes.
2. Compare the TARGET center to the CENTRAL center along each of
{axis}:
      • "+" if TARGET lies in the positive direction
      • "{" if in the negative direction
      • "0" if approximately equal (centers aligned along that axis)
3. Wrap your full step-by-step reasoning in <think>...</think>.
4. Then emit exactly one line, wrapped in <answer>...</answer>,
listing only
    axes {axis} with their sign or 0.
Do **not** include any extra text or prose.
```

```
% VLM USER PROMPT
CENTRAL: {central}
TARGET:  {target}

Return exactly:
<think>...step-by-step reasoning...</think>
<answer>(±X, ±Y, ±Z)</answer>

Valid examples:
  <think>I see red and green axes...</think>
  <answer>(+X, -Y)</answer>

  <think>Blue axis only is clear...</think>
  <answer>(-Z)</answer>
```

LLM PLANNER MODULE PROMPTS

```
% LLM SYSTEM PROMPT
You control a camera in a 3D scene. Your goal is to decide the
signs (+,0,-)
of TARGET relative to CENTRAL on axes X,Y,Z by choosing successive
viewpoints.

**Camera Coordinate System:**
- Azimuth 0°: X-axis points towards viewer, Y-axis points right
- As azimuth increases (clockwise rotation):
  - Azimuth 90°: Y-axis points towards viewer, X-axis points left
  - Azimuth 180°: X-axis points away from viewer, Y-axis points
  left
  - Azimuth 270°: Y-axis points away from viewer, X-axis points
  right
- Elevation 0°: Camera views from directly above (top-down)
- Elevation 90°: Camera views from horizontal level
- Elevation 180°: Camera views from directly below (bottom-up)

On every turn you will receive:
- threshold \tau (a float in [0,1])
- belief_state:
    {
      "X": {"+" : p_plus, "0": p_zero, "-" : p_minus},
      "Y": {...},
      "Z": {...}
    }
- history: a list of previously checked views, each entry:
    {
      "view":   {"az": az_deg, "el": el_deg},
      "answer": "(±X, ±Y, ±Z)" or shorter,
      "confidence": {"X":cX, "Y":cY, "Z":cZ}
    }

If **all** axes have max(belief) \geq \tau, you should stop.
Otherwise choose the
next best view. Note that you can revisit some views to stengthen
your belief.

You should also decide which axes you want to focus on in a view.
For example,
if you choose a view that shows the XY plane, then you should focus
on only
the X axis and Y axis or even just focus on X or Y axis.

Note that the confidence score represents the reliability of the
answer got from
that view. Zero confidence score for a view indicates that the
relative
position is not clear revealed through that view.

**Rules**
1. Wrap your internal reasoning in <think>...</think>
2. Then emit exactly one <answer>...</answer> containing **only**
this JSON:

{
  "action":   "CAPTURE"|"STOP",
  "view":     {"az": <number>, "el": <number>} | null,
  "axis":     ["X", "Y"]
}
No extra text or fields.
```

```
% LLM FIRST TURN PROMPT
# First turn (no belief_state or history)
Task: find (±X,±Y,±Z) for TARGET={target} vs CENTRAL={central}.
Threshold \tau = {tau}.

Propose your initial viewpoint.
Respond with:
<think>...</think>
<answer>{{
  "action": "CAPTURE",
  "view": {{"az": <num>, "el": <num>}},
  "axis": ["axes to focus on for this view"]
}}</answer>

% LLM INTERMEDIATE TURN PROMPT
# Subsequent turn
Threshold \tau = {tau}
belief_state = {belief_state}
history      = {history}

Decide whether to STOP or pick another view.
Respond with:
<think>...</think>
<answer>{{
  "action":   "CAPTURE"|"STOP",
  "view":     {{"az": <num>, "el": <num>}} | null,
  "axis": ["axes to focus on for this view"] | null
}}</answer>
```

## A.5   EXAMPLES OF MORE SPATIAL TASKS

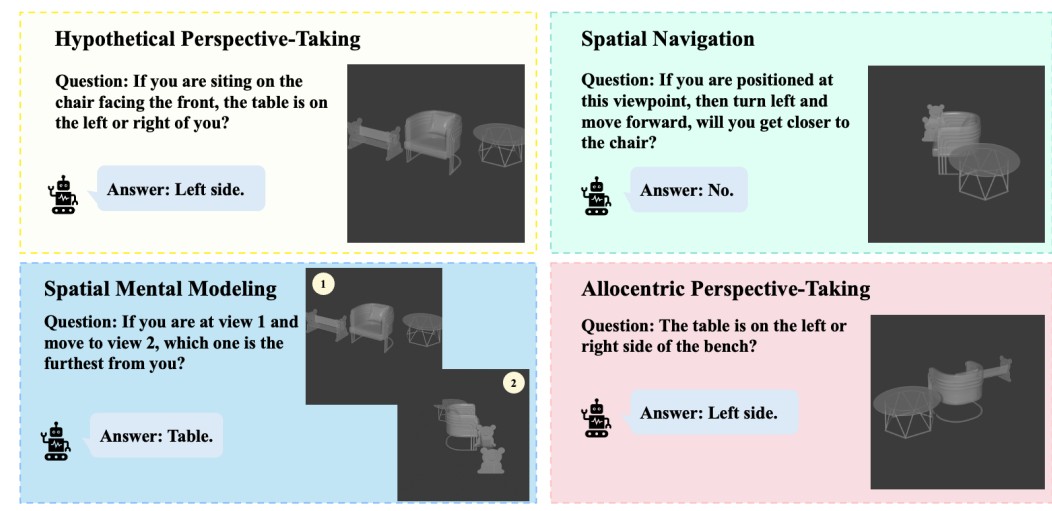

Figure 9: Examples of more spatial tasks that can be created using our data generation pipeline.

## A.6 FAILURE CASES

In this section, we present a glimpse of failure cases we observed during the evaluations.

### A.6.1 3D DoF=3 (CLAUDE 3.7 SONNET)

In this challenging 3D spatial reasoning task, the model correctly identifies the relationships on the X and Y axes but fails to accurately determine the relative position on the Z-axis.

- **Ground Truth:** (-X, +Y, +Z)
- **Model's Answer:** (-X, +Y, 0Z)

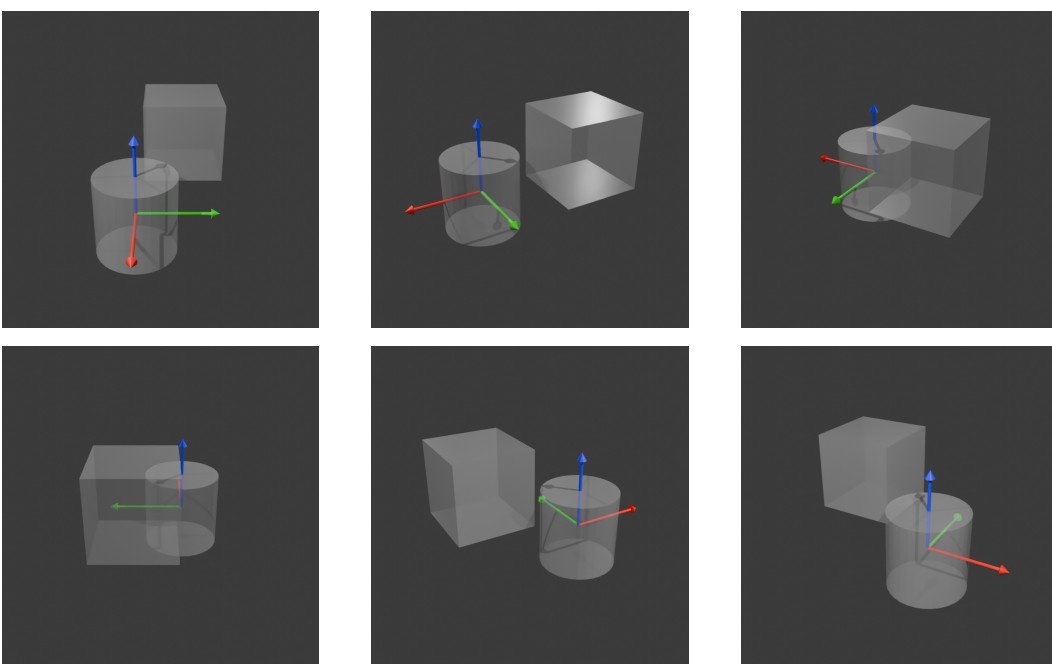

Figure 10: Six isometric views provided to the model for the 3D DoF=3 task.

*Reasoning and Answer:*

Looking at the different viewpoints of the 3D scene, I can analyze the relative position of the cube compared to the cylinder.
From the images, I can see that:

- X-axis: The cube appears to be positioned mainly in the negative X direction relative to the cylinder (to the left in several views)
- Y-axis: The cube appears to be positioned in the positive Y direction relative to the cylinder (forward in several views)
- Z-axis: The cube and cylinder appear to be at approximately the **same height** on **the Z-axis**

<answer>(-X, +Y, **0Z**)</answer>

### A.6.2 2D COLORS (CLAUDE 4 SONNET)

In this task, the model is provided with three canonical 2D views (front, side, and top). The model demonstrates a classic failure in view integration. It correctly processes the x-axis relationship (-X), which is salient in the top and front views. However, when trying to determine depth (Y-axis from the side/top views) and height (Z-axis from the front/side views), it incorrectly concludes the objects are at the "same level."

- **Ground Truth:** `(-X, +Y, +Z)`
- **Model's Answer:** `(-X, 0Y, 0Z)`

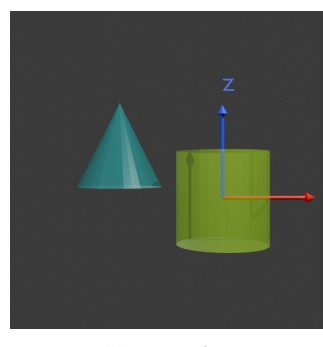 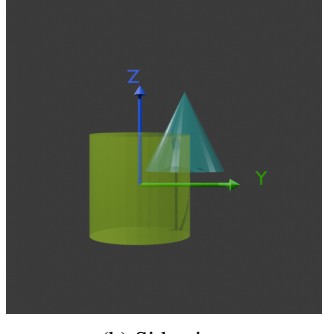 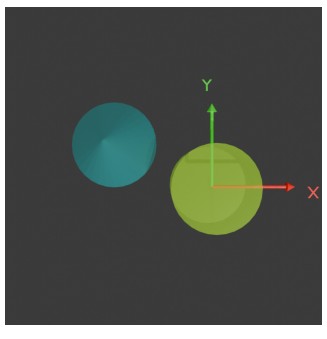

| (a) Front view | (b) Side view | (c) Top view |

Figure 11: Example task of 2D three view with colors.

***Reasoning and Answer:***

Based on examining the 3D scene from multiple viewpoints, I can determine the relative position of the teal cone compared to the lime cylinder.
Looking at all three viewpoints:

- X-axis: The cone appears to be to the left of the cylinder (negative X direction)

- Y-axis: The cone appears to be at approximately the **same level** as the cylinder on the Y-axis

- Z-axis: The cone appears to be at approximately the **same height** as the cylinder on the Z-axis

`<answer>(-X, `**`0Y, 0Z`**`)</answer>`

### A.6.3 ViewNavigator (GPT-5): VLM Perception Failure

This case illustrates a failure where the agent incorrectly determines the position on the Y-axis, leading to an incorrect final answer. The failure stems from a critical misperception by the VLM, which then misleads the LLM planner. This single error propagates through the agent's belief state, causing the LLM planner to develop confirmation bias and eventually accept the wrong conclusion.

- **Ground Truth:** `(-X, -Y, +Z)`
- **Agent's Final Answer:** `(-X, 0Y, +Z)`

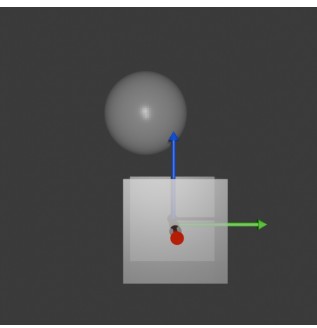

Figure 12: Image observed by the VLM from a side view (`az=0, el=90`).

***Reasoning and Anwser:***

> `<think>`Identify axes: green is +Y to the right; blue is +Z upward. Compare centers: the sphere is directly above the cube with **no horizontal (Y) offset**, so along Y the centers align (0). Along Z, the sphere lies above the cube center, so it is in the positive Z direction (+).`</think>` `<answer>`(**0Y**, +Z)`</answer>`

### A.6.4   VIEWNAVIGATOR (GEMINI 2.5 FLASH): LLM PLANNING FAILURE

This case demonstrates a fundamental failure in the LLM's spatial reasoning, where there is a disconnect between its stated goal and its proposed action. It correctly identifies the need for a view of the XZ plane but incorrectly believes that an elevation of 90° will achieve this. A view with an elevation of 90° is a top-down view, which clearly displays the XY plane while making it impossible to accurately judge relative positions along the X-axis. The model's stated goal is contradicted by its proposed action, highlighting a limited capacity to translate a desired outcome into the correct geometric parameters.

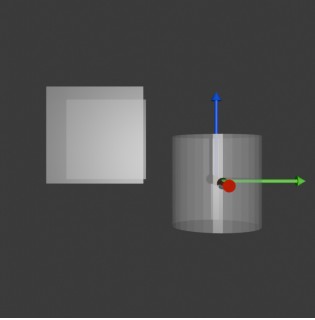

Figure 13: The view generated by the LLM's proposal (`az=0, el=90`).

***Reasoning and Planning:***

> `<think>` ... A view with azimuth 0° and elevation 90° would provide a side view focusing on the XZ plane, which might help clarify the X axis. This view also gives us a chance to improve our understanding of the Z axis. Therefore, I will choose a view with azimuth 0° and elevation 90°, focusing on the X and Z axes. `</think> <answer>` "action": "CAPTURE", "view": "az": 0, "el": 90, "axis": ["X", "Z"] `</answer>`

