# OpenReview forum: "Can VLMs Reason Through Multiple Views?"
_ICLR.cc/2026/Conference — ICLR 2026 Conference Desk Rejected Submission_

### Official Review · Reviewer_rgfn · 2025-10-20

**Soundness:** 3
**Presentation:** 3
**Contribution:** 3
**Rating:** 6
**Confidence:** 3

**Summary:**

This paper introduces a new benchmark to test the spatial reasoning of Vision Language Models (VLMs).  The benchmark is deliberately designed to test the ability to VLMs to utilize information from multiple views.  The authors use the new benchmark to test a variety of frontier VLMs and identify common failure modes.  They also propose an agentic framework for selecting informative views, which is shown to give a large performance for all the VLMs tested.

**Strengths:**

This task seems very useful for any workflows related to the assembly of individual assets or components in a virtual environment.  The benchmark will be useful for the community.

The analysis of failure modes was interesting and useful.

In addition to identifying problems, the authors propose the ViewNavigator system which clearly helps boost performance.

**Weaknesses:**

I couldn’t see any discussion for how the object poses were chosen.  Section 4.2 discusses only the relative placement (translation) of the objects, but not their pose (rotation).  Looking at Figure 2 I’m expecting the pose of the chair to be facing the table.  Using this semantic information, I can guess the relative placement of the two objects.   It feels like the benchmark is designed to check that the VLM doesn’t make this kind of assumption.

The additional cost of using the ViewNavigator is not discussed.  While it clearly provides great benefit, it would be useful to weigh this against the additional cost.

**Questions:**

How much did it cost to get the observed performance boost using the ViewNavigator?  Figure 7b is a nice comparison of the performance for a base method and  ViewNavigator.  Knowing the relative cost for each VLM would be useful.

---

> ### Author Response · Authors · 2025-11-29
>
> 1. We confirm that for real-world objects, we utilized the canonical pose provided by the source dataset (3DCoMPaT++) without introducing random rotations (rotation is fixed in the original furniture dataset). We acknowledge this was an oversight in the writing of Section 4.2. The diagnostic integrity of MVBench is preserved because the relative position (translation) of the target and central objects is fully randomized. This randomization ensures that the relative placement is equally probable in every direction, effectively decoupling the geometric task from any potential semantic bias a VLM might have (e.g., assuming a chair must face a table). Furthermore, our dataset is not limited to pairs like chair/table but includes diverse, less semantically constrained objects (e.g., sofa, bed, generic geometric primitives like cubes and cylinders), which further reduces the likelihood of VLMs succeeding or failing based on simple semantic correlation, keeping the focus on multi-view reasoning.
>
> 2. ViewNavigator actively checks approximately 8 viewpoints on average before confidently terminating the task. Given that the framework uses 5 jittered images per selected viewpoint to assess stability, this results in an average cost of approximately 40 images per task compared to 6 images used in main evaluations. We also tested that simply scaling up the number of images does not enhance VLMs’ performance. We evaluated with 10 viewpoints for the base models and saw basically no improvements. We would like to add this figure in our camera-ready version. Our intention is not to provide the ultimate solution but to demonstrate that such a simple scaffold shows promising potential to enhance VLM performance. The significant performance gains validate the utility of an active perception paradigm and encourage future research into more sophisticated, potentially fine-tuned or geometry-aware, agentic frameworks.

---

### Official Review · Reviewer_Jiec · 2025-10-29

**Soundness:** 3
**Presentation:** 2
**Contribution:** 2
**Rating:** 2
**Confidence:** 4

**Summary:**

This paper presents MVBench, a benchmark expressly designed to evaluate a VLM's ability to integrate information from multiple viewpoints for 3D scene understanding. A key contribution is the benchmark's extensible data-generation pipeline, which allows researchers to procedurally create new scenes with varied 3D assets and camera setups. An evaluation of several frontier VLMs on MVBench reveals consistent failure modes: while models can handle 2D planar relations, they exhibit marked difficulty with 3D spatial relations and aggregating information across views. The study also uncovers strong inductive biases, such as a reliance on conventional coordinate axis orientations. To address these shortcomings, the authors propose ViewNavigator, a multi-agent framework that actively selects informative viewpoints and fuses multi-view evidence via belief-updating. This framework is shown to improve the performance of base models on MVBench by over 50%.

**Strengths:**

1. A contribution is the proposed benchmark is not a static dataset. It allows researchers to construct VQA using different 3D assets, which has further room to be extended in the future.

2. The analysis is thorough, moving beyond simple accuracy to uncover further failure patterns. It reveals that VLMs excel at 2D planar relations but fail to aggregate this information into a coherent 3D model

3. The paper presents an interesting motivation and explain why such benchmark would be valuable.

**Weaknesses:**

1. As a paper where the new benchmark is the core contribution, the paper fails to explain the dataset composition clearly, which includes how many data is created, how many objects are typically involved in a scene and so on. Also, the entire pipeline is synthetic and all images are rendered in Blender. The benchmark does not test the ability of models to handle the complex lighting, textures, and occlusions of real multi-view 3D scenes.

2. The benchmark claims it is targeted at 3D scene comprehension, but the only task evaluated is determining the relative position of a target object from a central object. This is a limited, low-level proxy for the high-level reasoning, like part assembly, that the paper uses as its core motivation.

3. For a benchmark, the evaluation set is small if I understand it correctly. The paper reports using 100 tasks per task variant for the main evaluation and only 50 tasks for the ViewNavigator experiments. If each task is one VQA pairs, the small evaluation set may not be robust and informative enough to draw conclusions from.

4. The ViewNavigator agent is relatively simple. It is heuristic and based on jittering views and vote-counting. The LLM planner's job is reduced to resolving uncertainty on a simple 3-vector.

**Questions:**

Please refer to the weakness session.

---

> ### Author Response · Authors · 2025-11-29
>
> 1. The reviewer is concerned about the lack of dataset details, the simplicity of the scenes, and the heavy reliance on synthetic data.
> - Dataset Composition Details: We apologize if the location of the details was unclear. The dataset composition and size are explicitly detailed in Section 4.2 and the Reproducibility Statement. Each scene involves two objects (one central, one target). The benchmark utilizes six viewpoints with uniformly distributed azimuth angles for the main evaluation
> - We emphasize that the highly controlled synthetic scenes are a deliberate design choice to isolate the core cognitive failure: 3D reasoning and multi-view integration. Also, our benchmark serves as a diagnostic test for VLMs to operate 3D software like Blender. We intentionally test whether VLMs can interpret a fixed, view-independent global coordinate system and aggregate information across views in a 3D modeling environment.
>
> 2. We argue that the task is not merely a proxy but the most atomic prerequisite for high-level reasoning. Accurately determining the relative spatial relationship of parts (e.g., Is the leg in the positive-Z or negative-Z direction relative to the table top?) is the atomic step required for planning the assembly sequence and execution. The simple diagnostic format is the foundation for complex task execution.
>
> 3. We respectfully maintain that the evaluation set size is sufficient for statistical significance given the benchmark's purely diagnostic nature and all tasks are of the same fundamental type. Crucially, to ensure robustness, we ran all VLM evaluations on 100 tasks per task variant and repeated each experiment three times, reporting the averaged results. For ViewNavigator, due to the computational cost, we used 50 tasks per task variant and repeated them three times as well. This methodology yields statistically reliable and robust conclusions.
>
> 4. We acknowledge that ViewNavigator is designed to be a minimalist, plug-and-play scaffolding that operates without requiring post-training or external geometry-based image analysis. Our intention was not to propose an ultimate solution, but to demonstrate that even a relatively simple scaffold is highly effective. This demonstrates a promising path forward for mitigating the observed weaknesses in multi-view integration. The significant performance gains achieved by ViewNavigator serve as a compelling existence proof. We hope this framework encourages future research on model finetuning and the development of more sophisticated, geometry-aware agentic systems.

---

### Official Review · Reviewer_qFhw · 2025-10-31

**Soundness:** 1
**Presentation:** 2
**Contribution:** 1
**Rating:** 2
**Confidence:** 4

**Summary:**

This paper introduces MVBench, a benchmark designed to specifically test VLMs on their ability to perform multi-view integration for holistic 3D scene understanding. The core task involves inferring the 3D relative position of two objects from a set of rendered images. The authors identify key failure modes in current frontier close-sourced VLMs concerning 3D spatial reasoning and multi-view integration, leading to the proposal of ViewNavigator, an active planning, multi-agent framework designed to boost performance on the benchmark.

**Strengths:**

1. The paper's structure is clear, progressing from the motivation behind the problem to the construction of the benchmark, and finally to the testing and analysis of the results.
2. The paper includes rich charts, attempting to analyze model performance from multiple aspects.

**Weaknesses:**

1. The visual quality of the synthesized images is poor. The geometries used are simple and limited, and the rendering is far from photo-realistic, making it distant from real-world application domains. Overall, the data collection method is simple and direct, lacking significant technical difficulty or contribution.
2. The question format is monotonous, and it lacks obvious application-level significance. It's unclear what practical value this specific format provides beyond a diagnostic label. Basic information about the benchmark size, such as the total number of Q&A pairs, is not clearly stated.
3. The evaluation primarily relies on testing commercial, closed-source model APIs, with a lack of testing on other open-source or academic VLM architectures, which limits its practical guidance for VLM developers.
4. The proposed ViewNavigator, presented as a major contribution, is only allocated a very small section of the paper, and the description of its modules, workflow, and underlying mechanisms is insufficiently detailed.

**Questions:**

1. What is the specific positioning of this benchmark regarding difficulty and the problem it aims to solve? Compared to the numerous existing VLM benchmarks, where do the unique value and distinctiveness of MVBench lie?
2. Given the finding that performance drastically improves when breaking down the task into 2D single-view prediction, does the primary value of MVBench reside in diagnosing the failure of 3D integration, rather than being a challenging task?

---

> ### Author Response · Authors · 2025-11-29
>
> 1. We respectfully emphasize that our benchmark is primarily designed as a diagnostic benchmark, not a photorealistic scene understanding task. Our intentional use of clean, synthetic scenes with clearly marked axes is crucial for the task: testing a VLM's ability to interpret a fixed global coordinate system and aggregate information across views. This is analogous to a VLM operating within a 3D modeling environment like Blender and SolidWorks, where precise geometric reasoning is required over visual fidelity. We highlight the extensibility of our data generation pipeline as a core technical contribution. The modularity allows the research community to easily generate new datasets tailored to specific tasks; examples of extending to tasks proposed in other benchmarks are shown in Appendix A.5. Our pipeline supports adding realistic layouts. This can be easily achieved by importing realistic backgrounds while rendering. But adding realistic layouts deviates our main diagnostic intentions.
>
> 2. Inferring relative position using a global coordinate system is a non-negotiable prerequisite for any VLM deployed as an agent in domains like mechanical engineering or 3D modeling. Our task format is not monotonous but rather a rigid, diagnostic interface.
> - Application Significance: The task is directly motivated by the furniture assembly example discussed in Section 3. Accurately determining the relative spatial relationship of parts (e.g., Is the leg in the positive-Z or negative-Z direction relative to the table top?) is the atomic step required for planning the assembly sequence and execution. The simple diagnostic format is the foundation for complex task execution.
> - Benchmark Size: We confirm the total dataset size is clearly stated in the Appendix. We used 100 tasks per task variant and repeated each experiment three times, ensuring statistical significance for a diagnostic benchmark.
> Our primary goal is to benchmark the capabilities of the most powerful, frontier VLMs currently available (GPT-5, GPT-4o, Gemini 2.5, Claude 4) to define the current state-of-the-art ceiling for this challenging task. Given that these frontier closed-source models like GPT-4o and Claude-4-Sonnet only achieve around random chance performance, it is a reasonable inference that smaller, less capable open-source models would perform worse. The benchmark serves to highlight that multi-view integration is a fundamental, unsolved problem even for the largest models.
>
> 3. We acknowledge that, due to strict page limits, we moved the comprehensive technical details of ViewNavigator to the Appendix. We confirm that Appendix A.3 provides a detailed breakdown of the Belief State and Update mechanism, including the Dirichlet distribution model and the formulas for the Wilson Lower Bound Score and Relative Entropy Score. Furthermore, Appendix A.4 provides the hyperparameters and detailed LLM Planner and VLM Perception module prompts.
>
> Question 1: MVBench is positioned as a fundamentally diagnostic benchmark. The unique value and distinctiveness of MVBench lie in its explicit focus on multi-view information integration for unified 3D understanding, framed specifically within a 3D modeling software interface setting. This addresses a gap left by existing benchmarks.
> - MVBench is the first benchmark designed to test VLMs' ability to visually understand and reason using a fixed global coordinate system (X, Y, Z axes), which is fundamental to any 3D modeling software like Blender. This setup serves as a prerequisite test for building VLM-based agents for applications like 3D model design and mechanical engineering.
> - The task is challenging because it requires VLMs to mentally construct a coherent 3D model of the scene based on partial, sequential 2D observations and anchor this mental model to a viewpoint-independent global coordinate system. The core problem is not visual complexity but cognitive integration complexity (i.e., can the VLM build and reason over an internal 3D state?).
>
> Question 2: The finding that performance improves when breaking the task into 2D single-view prediction reinforces, rather than diminishes, the value of MVBench.
> - This provides the critical diagnostic insight that VLMs struggle with two independent failure modes: 3D Perception/Reasoning and Multi-View Integration as we discussed in section 5.
> - Even after simplifying the task to 2D multi-view reasoning, the performance of models (like GPT-4o, Claude 4 Sonnet, and Gemini 2.5 Flash) remains low, indicating that even the sub-tasks are challenging for VLMs and that MVBench remains a rigorous test of necessary spatial cognition. The low accuracy, far below human performance, indicates that multi-view integration is a fundamental, unsolved problem even for the largest models.

---

### Official Review · Reviewer_W6cC · 2025-11-01

**Soundness:** 2
**Presentation:** 2
**Contribution:** 2
**Rating:** 2
**Confidence:** 3

**Summary:**

The paper introduces MVBench, a benchmark designed to test whether VLMs can fuse information from multiple views to form a coherent 3D understanding, along with a framework for generating scenes to scale the benchmark. Evaluating frontier VLMs on MVBench, authors find failure patterns that models struggle with 3D spatial relations and aggregating information across views, and they show biases to axis conventions and color/texture variations. To mitigate these issues, the paper proposes ViewNavigator, a multi-agent framework that selects informative viewpoints and fuses multi-view evidence via belief updating, yielding large gains over base models on MVBench.

**Strengths:**

* Presents a new novel dataset that is tailored for measuring multi-view-based reasoning ability of VLMs, along with detailed descriptions on the dataset construction process.

* Delivers specific failure pattern analysis of state-of-the-art VLMs based on the proposed benchmark, offering new insights on their limitation on multi-view inputs.

* Along with the evaluation framework, authors further suggest a solution agentic system that performs accurate spatial reasoning based on multi-view observations.

**Weaknesses:**

* Overall, while the paper addresses an important problem of **multi-view reasoning** for VLMs, the proposed benchmark and method are highly task-specific and do not convincingly demonstrate generalized multi-view reasoning ability. Given that prior benchmarks already cover multi-view reasoning, using both synthetic and real images, the contribution of this work is unclear.

* There are multiple missining citations and discussions regarding closely related benchmarks on multi-view spatial reasoning. Discussions on the novely of MVBench in comparison to these benchmark should be provided. Benchmarks like **SITE [1], MM-Spatial [2], and SPAR-Bench [3]** include a rich set of multi-view QA tasks to asses multi-view reasoning of VLMs. Moreover, these works also include real-image data.

* From the examples, the dataset appears to be a trivial, synthetic collection of tasks, raising concerns about whether it truly assesses VLM's ability to reason over multi-view inputs in the wild. Given that several prior benchmarks already provide principled data-generation pipelines for multi-view reasoning with real-world scenes (e.g., MindCube [4], SPAR-Bench [3], MM-Spatial [2]), the motivation and technical contribution of introducing a much simpler dataset are unclear.

* The connection between the **furniture part–assembly task** and real-world multi-view reasoning is unclear. The task appears overly specialized rather than general, and in many cases part assembly seems solvable with a single view. Could the authors clarify why a multi-view formulation is necessary in this case?

* **ViewNavigator** may address the specific problem posed, but it is unclear whether it improves the underlying VLM’s intrinsic spatial reasoning. While the paper’s main goal is to diagnose limitations in multi-view reasoning, the ViewNavigator framework primarily builds an agentic system that compensates for those limitations. There seems to be a misalignment between the problem definition and solution in the paper.

---

[1] SITE: towards Spatial Intelligence Thorough Evaluation, Wang et al., ICCV 2025

[2] MM-Spatial: Exploring 3D Spatial Understanding in Multimodal LLMs, Dexberger et al., ICCV 2025

[3] From Flatland to Space: Teaching Vision-Language Models to Perceive and Reason in 3D, Zhang et al., NeurIPS 2025

[4] Spatial Mental Modeling from Limited Views, Yin et al., 2025

**Questions:**

* While the paper focuses on synthetic rendering setups, could the proposed framework be extended to more realistic data synthesis? For instance, adding a room layout and placing multiple object and rendering from multiple viewpoints could extend MVBench to include more realistic images.

* Does fine-tuning VLMs on the proposed benchmark lead to improvement in VLMs' mutli-view reasoning? I believe this would clarify whether the proposed benchmark is a reasonable task or not. For reference, multiple spatial reasoning benchmark works show that their data indeed improves the base VLMs through fine-tuning: SAT [1], MindCube [2], SPAR-Bench [3], RoboSpatial [4].

---

[1] SAT: Dynamic Spatial Aptitude Training for Multimodal Language Models, Ray et al., COLM 2025

[2] Spatial Mental Modeling from Limited Views, Yin et al., 2025

[3] From Flatland to Space: Teaching Vision-Language Models to Perceive and Reason in 3D, Zhang et al., NeurIPS 2025

[4] RoboSpatial: Teaching Spatial Understanding to 2D and 3D Vision-Language Models for Robotics, Song et al., CVPR 2025

**Details Of Ethics Concerns:**

No concerns.

---

> ### Author Response · Authors · 2025-11-29
>
> 1. We thank you for providing these valuable references. Our benchmark explored a foundational gap in VLMs' ability to integrate fragmented 2D views into a coherent, world-centric 3D representation using a fixed, view-independent global coordinate system. This differs from the cited works, which focus on testing local, viewpoint-dependent queries. And just as our data generation pipeline can generate these viewpoint-dependent tasks (Appendix A.5), this global multi-view reasoning tests the union of those viewpoint-dependent abilities. Compared to existing VLM multi-view benchmarks that emphasize complex, realistic scenes, our benchmark intentionally uses simple, synthetic scenes to isolate and diagnose fundamental failures in multi-view integration. This simplicity ensures the benchmark serves as a "fundamental diagnostic" role. It acts as a prerequisite VLMs must pass before deployment in precision-critical domains like mechanical engineering, where understanding absolute coordinates, such as in Blender-like GUIs, is essential.
>
> 2. The synthetic design is deliberate for diagnostic purposes. We aim to expose how poorly current VLMs handle even basic multi-view 3D reasoning in a clean environment, not to create overly complex scenes that might confound failures with extraneous factors, such as occlusions or lighting in real images. Existing benchmarks like MindCube, SPAR-Bench, and MM-Spatial provide valuable principled pipelines for realistic multi-view data, but MVBench's extensible pipeline focuses on controlled, fundamental tasks with a well-defined coordinate system. This aligns with our goal of testing core abilities required in mechanical engineering. Also we would like to point out that these benchmarks are  concurrent works that were published while we were finishing up our work. We would like to add these citations in our camera-ready version.
>
> 3. The furniture assembly motivation in Section 3 illustrates a practical downstream task where multi-view reasoning is essential, not optional. Single-view observations often lead to misconceptions, such as hidden misalignments in non-convex parts as shown in Figure 1, and bounding-box descriptions fail for irregular shapes. We clarify that while some assemblies might seem single-view solvable, real non-convex parts demand multi-view to resolve ambiguities.
>
> 4. We acknowledge that ViewNavigator is designed to be a minimalist scaffolding that operates without enhancing VLM’s intrinsic spatial reasoning. Our intention was not to propose an ultimate solution, but to demonstrate that even a relatively simple scaffold is highly effective. This demonstrates a promising path forward for mitigating the observed weaknesses in multi-view integration. The significant performance gains achieved by ViewNavigator serve as a compelling existence proof. We hope this framework encourages future research on model finetuning and the development of more sophisticated agentic systems.
>
> - Question 1: Extension to Realistic Synthesis: Yes, our pipeline supports adding realistic layouts. This can be easily achieved by importing realistic backgrounds while rendering. But adding realistic layouts deviates our main diagnostic intentions.
> - Question 2: Fine-tuning is out of scope of our main diagnostic intention and this is left as future work. We would like to explore whether our dataset can be used to improve the multi-view reasoning and train VLMs to actually master the furniture assembly tasks in the future.

---

### Note · Program_Chairs · 2026-01-17
**Submission Desk Rejected by Program Chairs**

The following references in this submission do not refer to real documents and/or have major errors in bibliographic information:

 Mark G. Stokes et al. The prefrontal cortex and cognitive control. Annual Review of Neuroscience, $44: 403-423,2021$